# A High-Affinity Monoclonal Antibody Against the Pancreatic Ductal Adenocarcinoma Target, Anterior Gradient-2 (AGR2/PDIA17)

**DOI:** 10.3390/antib13040101

**Published:** 2024-12-05

**Authors:** Reeder M. Robinson, Leticia Reyes, Benjamin N. Christopher, Ravyn M. Duncan, Rachel A. Burge, Julie Siegel, Patrick Nasarre, Pingping Wang, John P. O’Bryan, G. Aaron Hobbs, Nancy Klauber-DeMore, Nathan G. Dolloff

**Affiliations:** 1Department of Pharmacology and Immunology, Medical University of South Carolina, Charleston, SC 29425, USA; robinree@musc.edu (R.M.R.); reyesl@musc.edu (L.R.); christob@musc.edu (B.N.C.); rduncan@mplnet.com (R.M.D.); 2Department of Biochemistry and Molecular Biology, Medical University of South Carolina, Charleston, SC 29425, USA; burgera@musc.edu (R.A.B.); obryanjo@musc.edu (J.P.O.); hobbsg@musc.edu (G.A.H.); 3Department of Surgery, Medical University of South Carolina, Charleston, SC 29425, USA; julie.siegel@louisville.edu (J.S.); nasarre@musc.edu (P.N.); demore@musc.edu (N.K.-D.); 4Creative Biolabs, Inc., Shirley, NY 11967, USA; pwang@creative-biolabs.com; 5MUSC Hollings Cancer Center, Charleston, SC 29425, USA; 6Zucker Institute for Innovation Commercialization, Charleston, SC 29425, USA

**Keywords:** AGR2, PDIA17, anterior gradient-2, protein disulfide isomerase, pancreatic cancer, PDAC, LYPD3, SMAD4, monoclonal antibody, scFv

## Abstract

Background/Objectives: Anterior Gradient-2 (AGR2/PDIA17) is a member of the protein disulfide isomerase (PDI) family of oxidoreductases. AGR2 is up-regulated in several solid tumors, including pancreatic ductal adenocarcinoma (PDAC). Given the dire need for new therapeutic options for PDAC patients, we investigated the expression and function of AGR2 in PDAC and developed a novel series of affinity-matured AGR2-specific single-chain variable fragments (scFvs) and monoclonal antibodies. Results: We found that AGR2 was expressed in approximately 90% of PDAC but not normal pancreas biopsies, and the level of AGR2 expression correlated with increasing disease stage. AGR2 expression was inversely related to SMAD4 status in PDAC and colorectal cancer cell models and was secreted from cells into their media. In normal tissues, a high density of AGR2 was detected in the epithelium of cells in the digestive tract but was lacking in most other normal tissue systems. The addition of recombinant AGR2 to cell culture and genetic overexpression of AGR2 increased the adhesion, motility, and invasiveness of both human and mouse PDAC cells. Human phage display library screening led to the discovery of multiple AGR2-specific scFv clones that were affinity-matured to produce monoclonal antibody (MAb) clones with low picomolar binding affinity (S31R/A53F/Y). These high-affinity MAbs inhibited AGR2-mediated cell adhesion, migration, and binding to LYPD3, which is a putative cell surface binding partner of AGR2. Conclusions: Our study provides novel, high-affinity, fully human, anti-AGR2 MAbs that neutralize the pro-tumor effects of extracellular AGR2 in PDAC.

## 1. Introduction

Pancreatic cancer remains one of the deadliest cancers [1,2]. Pancreatic ductal adenocarcinoma (PDAC), the most common malignancy originating in the pancreas, is predicted to be the second most deadly form of cancer by 2030, behind only lung cancer [3]. Standard-of-care treatments lack efficacy and have remained largely unchanged for decades. These include surgical resection and adjuvant gemcitabine or FOLFIRINOX [4,5,6,7]. Surgery is the only path to a cure, although >50% of patients present with stage IV disease that has already metastasized [8], and for these patients, surgery is palliative. Newer agents, like the EGFR inhibitor erlotinib and albumin-bound paclitaxel (nab-paclitaxel), provide only marginal benefit [9,10,11,12]. As for immunotherapy, the PD-1 checkpoint inhibitor pembrolizumab/KEYTRUDA is effective against tumors with DNA mismatch repair (MMR) deficiency [13,14]; this molecular designation applies to <1% of PDAC patients [15]. Indeed, clinical trials have shown almost zero objective responses in PDAC patients enrolled in PD-1/PD-L1 checkpoint inhibitor clinical trials [16,17]. Thus, there is a pressing need for new therapeutic approaches and disease targets for the treatment of PDAC.

AGR2 (PDIA17) belongs to the protein disulfide isomerase (PDI) family, playing a pivotal role in protein folding by introducing and rearranging disulfide bonds in newly formed proteins. It shares homology with the Xenopus Anterior Gradient-2 (XAG2) gene, which is secreted by the cement gland in *Xenopus laevis,* where it orchestrates embryonic development by directing the polarization of the anterior dorsal ectoderm [18,19]. In humans, AGR2 expression is up-regulated in several cancers and is linked to increased progression and metastatic spread. Previous studies have reported that AGR2 was overexpressed in PDAC but not in normal pancreas or chronic pancreatitis tissue [20,21]. Studies have shown that AGR2 played a causal role in the initiation and progression of PDAC in mouse models [20], and a number of pro-oncogenic functions have been ascribed to it, including the promotion of cell survival, invasion, increased metastatic potential, and resistance to therapy [22,23,24]. AGR2 is up-regulated downstream of KRAS G12D mutation in PDAC [25] and is repressed by SMAD4 [26], which is lost by mutation or homozygous deletion in roughly 1/3 of PDAC patients [27]. It is an early biomarker of pancreatic intraepithelial neoplasia (PanIN) and PDAC and is secreted into blood serum and pancreatic fluid [28,29], as well as the cell culture supernatant of PDAC cell lines [30,31]. AGR2 is, therefore, an intriguing biomarker candidate, which others have proposed [32]. It is also a potential therapeutic target for PDAC, and while this hypothesis has not been thoroughly evaluated, our group and others have begun to explore this possibility [33,34].

AGR2 is an atypical member of the PDI family. Among the unique characteristics of AGR2 is its CXXS redox catalytic core [35], which differs from the quintessential CXXC thioredoxin-like domain of PDI isoforms. In addition, AGR2 expresses a C-terminal KTEL ER retention sequence, which has a reduced affinity for the KDEL receptor than the more common KDEL motif [36]. Perhaps not coincidentally, AGR2 has been reported to be secreted from cancer cells and found to be associated with the cell surface [37]. The function of AGR2 on the cell surface and in the extracellular space is not clear. It is thought to function as a chaperone that supports the folding, trafficking, and stability of cysteine-rich glycoproteins expressed on the cell membrane [31,32,38,39]. Its extracellular function has also been linked to increased migration and invasion of cancer cells, fibroblasts, and monocytes [40,41]. AGR2 expression has been closely tied to regulators of the TGFβ pathway, including TGFβ itself and SMAD4 [26,42], further implicating a role for AGR2 in cellular migration and, potentially, the epithelial-to-mesenchymal transition (EMT). Given the need to develop new therapeutics for PDAC and the substantial body of evidence implicating AGR2 in PDAC and other tumor types, we set out in this study to characterize AGR2 expression in primary PDAC samples and model systems, demonstrate its role in the motility of PDAC cells, and to develop a novel and high-affinity anti-AGR2 antibody that could be used for therapy.

## 2. Materials and Methods

*Cell Lines and Reagents:* Cell lines were purchased from the American Tissue Culture Collection (Manassas, VA, USA) and cultured according to the provider’s instructions. Cells were grown at 37 °C and 5% CO_2_ in the recommended base media supplemented with 10% heat-inactivated fetal bovine serum (FBS) (900-108, GeminiBio, Sacramento, CA, USA), and antibiotics (1% penicillin (10,000 units/mL), 1% streptomycin (10,000 µg/mL) were purchased from Selleck Chemicals (Houston, TX, USA).

*Protein extraction from pancreatic tumor tissues:* Human pancreatic tumor samples and their matched normal adjacent tissues from pancreatic resections were obtained from the MUSC Biorepository and Tissue Analyses Shared Resource. These de-identified tissues were obtained from institutional surgical specimens under an IRB-approved protocol and were stored at −80 °C prior to use. Samples were crushed in liquid nitrogen into a fine powder, and 30–100 mg were mixed in Radioimmunoprecipitation assay, or RIPA, buffer (#R3792; TEKnova, Hollister, CA, USA) for protein lysis. Lysates were sonicated and centrifuged at 13,000× *g*, and supernatants were collected. Lysates were dialyzed, and then an additional RIPA buffer was added. Lysates were used for Western blot, as described below.

*Subcloning:* Genes for AGR2, LYPD3, scFvs, and monoclonal antibodies were codon optimized for either *E. coli* or human cells by GenScript Biotech (Piscataway, NJ, USA. AGR2 was subcloned into the pET15TEV_NESG vector using the NdeI and BamHI restriction sites. All scFvs were subcloned into the pET22b vector using the NcoI and XhoI restriction sites. LYPD3 was subcloned into a custom pcDNA3.4 using NheI and XhoI with an N-terminal secretion signal and a C-terminal 6x-His tag. Monoclonal antibodies were subcloned into a custom pcDNA3.4 using NheI and AgeI with an N-terminal secretion signal. Proper ligation of each gene was confirmed by DNA sequencing at Eurofins Genomics (Louisville, KY, USA).

*AGR2 and scFvs Expression and Purification:* AGR2 (21–175) and all AGR2 scFvs were expressed in *E. coli* BL21(DE3) in LB medium. Protein expression was induced at OD_600_ ~0.6 with 400 µM IPTG at 18 °C overnight. Cells were harvested by centrifugation and resuspended in buffer A (25 mM HEPES, 300 mM NaCl, 25 mM imidazole, pH 7.5) with 1 mM PMSF and 50 µg/mL lysozyme using 3 mL of buffer per 1 g of cell paste. The mixture was stirred for 30 min to create a suspension, and the cells were then lysed with a Fisher Scientific Sonic Dismembrator Model 500 at 50% amplitude for 5 min at 4 °C (5 s pulses with 30 s delays). The insoluble material from the lysate was then removed by centrifugation at 48,400 × *g* for 1 h. The resulting supernatant was then loaded at 3 mL/min onto a 5 mL HisTrap FF crude column (Cytiva, Marlborough, MA, USA) equilibrated with buffer A. After loading, the column was washed with buffer A containing 75 mM imidazole, and the bound protein was subsequently eluted with buffer A containing 500 mM imidazole. AGR2 scFv were dialyzed overnight in PBS, pH 7.4, flash-frozen the following day in liquid nitrogen, and stored at −80 °C. AGR2 was dialyzed overnight in buffer A with no imidazole at 4 °C and 15 mg of 8x-His-TEV to liberate the N-terminal 6x-His tag. The following day, the protein was loaded at 3 mL/min onto a 5 mL HisTrap FF crude column (Cytiva, Marlborough, MA, USA) equilibrated with buffer A containing no imidazole, and the flow-through containing AGR2 (-) 6x-His tag was collected. AGR2 (-) 6x-His tag was then dialyzed in 50 mM HEPES, 150 mM NaCl, 1 mM DTT, 1 mM EDTA, 5% (*v*/*v*) glycerol, pH 7.5 overnight at 4 °C and flash frozen the following day in liquid nitrogen. Protein purity was determined using SDS-PAGE, and concentration was determined using A_280_.

*LYPD3 Expression and Purification:* LYPD3 (31–325) was expressed in expi293F cells following the manufacturer’s protocol. Briefly, 1 µg/mL pcDNA3.4 containing LYPD3 (31–325) with a secretion signal and C-terminal 6x-His tag was transfected into expi293F cells using expifectamine 293. Media was collected by centrifugation of cells 6 days later when cell viability was <60%. Media was then loaded at 5 mL/min onto a 5 mL HisTrap Excel column (Cytiva, Marlborough, MA, USA) equilibrated with buffer C (20 mM sodium phosphate, 500 mM NaCl, pH 7.4). After loading, the column was washed with buffer C containing 25 mM imidazole, and the bound protein was subsequently eluted with buffer C containing 500 mM imidazole. Eluted LYPD3 was then dialyzed overnight in PBS, pH 7.4 at 4 °C, flash-frozen the following day in liquid nitrogen, and stored at −80 °C. Protein purity was determined using SDS-PAGE, and concentration was determined using A_280_.

*Monoclonal Antibody Expression and Purification:* All monoclonal antibodies (MAbs) were expressed in expi293F cells following the manufacturer’s protocol. Briefly, pcDNA3.4 containing heavy chain and light chain with secretion signals were mixed to a 1:2 ratio and transfected with expifectamine 293 at 1 µg/mL total pcDNA3.4. Media was collected by centrifugation of cells 6 days later when cell viability was <60%. Media was then loaded at 1 mL/min onto a 1 mL rProteinA FF column (Cytiva, Marlborough, MA, USA) equilibrated with buffer D (20 mM sodium phosphate, pH 7.0). After loading, the column was washed with buffer D, and purified MAb was eluted with buffer E (100 mM sodium citrate, pH 3.0) into tubes containing neutralization buffer (1 M Tris-HCl, pH 9.0). Eluted MAbs were then dialyzed overnight in PBS, pH 7.4 at 4 °C, flash-frozen the following day in liquid nitrogen, and stored at −80 °C. Protein purity was determined using SDS-PAGE, and concentration was determined using A_280_.

*Phage Biopanning:* The phage display semi-synthetic human scFv library HuScl-2^TM^ (Creative Biolabs, Inc., Shirley, NY, USA) was used for screening against a recombinant human 6x-His tagged AGR2 protein. After four rounds of library biopanning against AGR2 protein immobilized on the surface of immunotubes, single phage clones were randomly picked and analyzed for phage ELISA against AGR2. The AGR2 binding clones were sanger-sequenced to identify the unique scFv clones. Further, scFv binding to the AGR2 protein was evaluated by soluble scFv ELISA using scFv produced by E. coli periplasmic expression, following published protocols [43].

*Antibody Affinity Maturation:* Based on the amino acid sequence of the parent anti-AGR2 scFv-15, VH and VL mutated-phage display scFv libraries were constructed via the error-prone PCR mutagenesis method [44]. In parallel, according to the sequence alignment and structural modeling analysis, a phage display semi-rational mutant scFv library via degenerate codon mutagenesis approach (NNK) was constructed [45]. The mutant scFv libraries were biopanned against biotinylated recombinant AGR2 protein using an in-solution selection method described previously [45]. In the screening system, an excess amount of parent MAb-15 protein was used for the competition to enrich phage clones with a slower k_off_ rate and, therefore, higher-binding affinity to AGR2 than the parent antibody. After 4 rounds of biopanning, single phage clones were picked for phage ELISA analysis against biotinylated AGR2 protein and further DNA sequenced for unique scFv sequences. The candidate scFv mutants were formatted into human IgG1 and further recombinantly expressed and purified by 293 cells and measured for affinity kinetics by ELISA.

*PDAC Spheroid Cultures:* To create KPC and KPC-AGR2 spheroids, 3000 cells were suspended in 30 µL of Matrigel (Corning, Tewksbury, MA, USA) and allowed to grow in 3D culture for 3 weeks before subsequent experiments. KPC-AGR2 spheroids were maintained in puromycin. For experimental assays, spheroids were suspended in TrypLE (Thermo Fisher Scientific, Waltham, MA, USA) to dissociate into single cells, then replated in Matrigel at 10,000 cells per 96-well plate in 30 µL of Matrigel. After allowing the gel to set, 150 µL of organoid medium was added. The next day, cells were treated with fresh medium containing antibodies at a final concentration of 1 µg/mL per well. After 14 days in an organoid medium with antibodies, images were taken using the ImageXpress^®^ Pico Automated Cell Imaging System (Molecular Devices, San Jose, CA, USA). At least 5 fields of view for each biological replicate were used to calculate the average spheroid size across experimental conditions using ImageJ. The medium used was composed of advanced DMEM/F12, 1× Pen/Strep, 10 mM HEPES, 1× Glutamax, 500 nM A83-01, 50 ng/mL mEGF, 1 mM N-acetylcysteine, 10 mM nicotinamide, 1× N2 supplement, 1× B27 supplement, and 50% mNoggin/R-spondin1/Wnt3A conditioned media.

*Western Blot:* Cells were collected on ice, rinsed with cold 1× PBS, centrifuged at 2500 rpm at 4 °C for 5 min, and lysed in 1× cell lysis buffer (Cell Signaling Technology, Danvers, MA, USA) with the addition of protease (Thermo Fisher Scientific, Waltham, MA, USA) and phosphatase (Thermo Fisher Scientific, Waltham, MA, USA) inhibitors. Cell lysates were clarified, and their relative protein concentrations were determined via Bradford assay using Bio-Rad Protein Assay Dye Reagent Concentrate diluted 1:5 with MilliQ water (Bio-Rad Laboratories, Hercules, CA, USA). Values were normalized to the lowest average absorbance at 595 nm. Gel samples were prepared by mixing lysates with SDS sample buffer containing β-mercaptoethanol (final concentration 1×) and 1× cell lysis buffer. Samples were boiled for 10 min, loaded on NuPAGE Bis-Tris Gel 4–12% (Invitrogen, Waltham, MA, USA), and subjected to gel electrophoresis at 55 mA for 1 h and 45 min in 1× NuPAGE MOPS SDS Running Buffer (Invitrogen, Waltham, MA, USA). Gels were transferred to polyvinylidene difluoride (PDVF) membranes at 300 mA for 2 h in 1× transfer buffer containing 25 mM trizma base, 192 mM glycine, and 20% methanol. PVDF membranes were blocked for 1 h at room temperature with 5% milk in TBS-Tween prior to incubation with primary antibodies in 5% milk TBS-Tween overnight at 4 °C. Primary antibodies were β-Actin (A5441, Sigma-Aldrich, St. Louis, MO, USA), AGR2 (13062, Cell Signaling Technology, Danvers, MA, USA), GAPDH (2118, Cell Signaling Technology, Danvers, MA, USA), SMAD4 (38454, Cell Signaling Technology, Danvers, MA, USA), PDIA1 (2446, Cell Signaling Technology, Danvers, MA, USA), Ran (sc-271376, Santa Cruz Biotechnology, Dallas, TX, USA). Membranes were washed with 1× TBS-Tween and subsequently incubated with secondary antibodies in 1% milk TBS-Tween for 2 h at room temperature. Secondary antibodies conjugated to horseradish peroxidase were goat anti-mouse IgG-H+L (31430, Invitrogen, Waltham, MA, USA) and goat anti-rabbit IgG-H+L. Detection was finalized using ECL (T32209, Thermo Fisher Scientific, Waltham, MA, USA) or Super Signal (34094, Thermo Fisher Scientific, Waltham, MA, USA) detection reagents.

*RT-PCR:* Total RNA was extracted from treated cells for quantitative reverse transcriptase polymerase chain reaction (RT-qPCR) using the Qiagen RNeasy Plus Mini Kit per the manufacturer’s instructions. Isolated RNA concentrations were determined via NanoDrop, and RNA was reverse-transcribed using the Luna Universal One-Step RT-qPCR Kit (New England BioLabs, Ipswich, MA, USA). Samples were run using the QuantStudio 3 Real-Time PCR System (Thermo Fisher Scientific, Waltham, MA, USA), and data were analyzed with QuantStudio3 qPCR Data Analysis. Primers: *AGR2* Fwd: 5′-ACAAAGGACTCTCGACCCAAA-3′, Rvs: 5′-GTGGGCACTCATCCAAGTGA-3′; *GAPDH* Fwd: 5′-ACAACTTTGGTATCGTGGAAGG-3′, Rvs: 5′-GCCATCACGCCACAGTTTC-3′.

*Tissue Micro Array (TMA) Immunohistochemistry:* Pancreas cancer (PA721a) and normal tissue (BN1021) TMA slides were acquired from TissueArray.com (accessed on 2 January 2019, Derwood, MD, USA). Slides were deparaffinized and rehydrated, and antigen retrieval was performed using heat-mediated, citrate-based Antigen Unmasking Solution (3300-250, Vector Laboratories, Newark, CA, USA) for 30 min. Endogenous peroxidases were blocked with hydrogen peroxide at a concentration of 1:100 (#5155-01, J.T. Baker, Phillipsburg, NJ, USA). Normal goat serum (#50062Z, Thermo Fisher) was used for non-specific protein blocking. Slides were incubated with the primary antibody at a 1:400 dilution (3062, Cell Signaling Technology, Danvers, MA, USA), and negative control was performed with no primary antibody for 90 min at room temperature (RT). Slides were subsequently incubated with the secondary antibody (biotinylated anti-rabbit, 1:150) for 30 min at RT, followed by a 30 min incubation with Vectastain (PK-4000, Vector Laboratories) and an incubation with DAB (1856090, Thermo Fisher Scientific, Waltham, MA, USA), followed by a hematoxylin counterstaining (H3401, Vector Laboratories, Newark, CA, USA). Slides were counterstained with fresh Weigert’s hematoxylin for 10 min. Tissues were then stained with Aniline Blue solution for 5 min, differentiated in acetic acid solution for 1 min, dehydrated, and cleared with xylene. Slides were washed and then dehydrated and cleared. The AGR2 staining percentage or Score (0–4) was determined by a blinded investigator.

*Adhesion Assays:* The 96-well MaxiSorp plates (Thermo Fisher Scientific, Waltham, MA, USA) were coated with 1 µg of AGR2 or BSA in coating buffer (50 mM sodium bicarbonate/carbonate, pH 9.6) and incubated overnight at 4 °C. The next day, plates were washed three times with PBS, pH 7.4, and antibodies were added to each well in 100 µL DMEM (ATCC, Manassas, VA, USA) supplemented with 10% fetal bovine serum and 1× antibiotics for 1 h at 37 °C. After incubation, 100 µL of KPC cells were added to each well and incubated for 1 h at 37 °C. After incubation, the medium was aspirated and washed with 100 µL of warmed non-supplemented DMEM and then with 100 µL of warmed PBS, pH 7.4. The PBS was then aspirated, and 50 µL of CellTiter-Glo (Promega, Madison, WI, USA) was added to each well for 5 min. Luminescence was recorded on a SpectraMax L Microplate Reader (Molecular Devices, San Jose, CA, USA) at 470 nm with a 1 s integration time.

*Migration Assays:* KPC cells were collected in PBS, pH 7.4, with 30 mM EDTA on ice for 30 min. Cells were resuspended in non-supplemented DMEM, and 3 × 10^5^ cells were added in 0.5 mL to Corning BioCoat Control Cell Culture Inserts (8 µm PET membrane). To 24 well plates, 0.75 mL of DMEM ± chemoattractant ± 500 ng/mL AGR2 or BSA were added to each well, and inserts containing cells were placed into their respective wells and allowed to incubate for 6 h at 37 °C with 5% CO_2_. After incubation, a cotton swab moistened with DMEM was used to gently scrape each well twice to remove any cells that did not migrate. The inserts were then flipped over and migrated cells on the bottoms of the inserts were fixed with 3.7% formaldehyde in PBS for 2 min. The bottoms of the inserts were washed with PBS twice, and cells were permeabilized with 100% methanol for 20 min. The bottoms of the inserts were then washed twice with PBS, and cells were stained with 1% toluidine blue in 1% borax for 15 min. The bottoms of the inserts were washed three times with PBS and once with dH_2_O, and the membranes were allowed to dry overnight. The next day, fixed and stained cells were counted manually under a microscope.

*Invasion Assays:* Corning BioCoat Control Cell Culture Inserts (8 µm PET membrane) were coated with 100 µL of Matrigel (300 ng/mL) in 10 mM Tris-HCl pH 8.0 + 0.7% NaCl for 3 h at 37 °C. After incubation, excess liquid that was not solidified was aspirated. KPC cells were collected in PBS, pH 7.4, with 30 mM EDTA on ice for 30 min. Cells were resuspended in 500 µL of non-supplemented DMEM, and 3 × 10^5^ cells were added to each transwell on top of the solidified Matrigel. To 24 well plates, 0.75 mL of DMEM ± chemoattractant ± 500 ng/mL AGR2 or BSA were added to each well, and inserts containing cells were placed into their respective wells and allowed to incubate overnight at 37 °C with 5% CO_2_. After incubation, liquid media was aspirated from the upper transwell, and the Matrigel was removed by taking a cotton swab moistened with DMEM and gently rubbing the well twice until the Matrigel was removed. The transwells were then transferred to fresh 24-well plates containing 200 µL of 0.25% trypsin, 2.21 mM EDTA, 1× sodium bicarbonate (Corning, Tewksbury, MA, USA), and incubated for 15 min at 37 °C. The transwells were removed, and trypsin was quenched with 20 µL of FBS. The liberated cells were then transferred to black 96-well plates, and 100 µL of CellTiter-Glo (Promega, Madison, WI, USA) was added to each well for 5 min. Luminescence was recorded on a SpectraMax L Microplate Reader (Molecular Devices, San Jose, CA, USA) at 470 nm with a 1 s integration time.

*ELISA Assays:* The 96-well MaxiSorp plates (Thermo Fisher Scientific, Waltham, MA, USA) were coated with 1 µg of AGR2, PDI, TXN, or BSA in coating buffer (50 mM sodium bicarbonate/carbonate, pH 9.6) and incubated overnight at 4 °C. The next day, plates were washed three times with washing buffer (PBS, 0.05% Tween 20, pH 8.0) and blocked with blocking buffer (50 mM Tris, 140 mM NaCl, 1% BSA, pH 8.0) for 30 min at room temperature. Plates were then washed three times with washing buffer, and varying LYPD3/scFv/MAb were incubated in diluent buffer (50 mM Tris, 140 mM NaCl, 1% BSA, 0.05% Tween 20 pH 8.0) for 1 h at 37 °C. Plates were then washed three times with washing buffer. For ELISA assays with varying LYPD3 and scFvs, anti-6x-His-peroxidase (I Invitrogen, Waltham, MA, USA) was added at a 1:10,000 dilution in the diluent buffer. For ELISA assays with varying MAbs, Protein L-peroxidase (Pierce Biotechnology, Waltham, MA, USA) was added at a 1:10,000 dilution in the diluent buffer. All secondary detection antibodies were allowed to incubate at room temperature for 1 h. Plates were then washed five times with washing buffer and subsequently developed using TMB substrate at room temperature for ~5 min. Development reactions were quenched with equal volumes of 180 mM sulfuric acid, and the absorbance at 450 nm was measured using a SpectraMax i3 plate reader (Molecular Devices, San Jose, CA, USA).

*Thermodynamic Stability Assay:* Thermodynamic stability of scFvs and MAbs was determined using a Tycho NanoTemper (Munich, Germany). Proteins were diluted to 200 µg/mL in PBS and drawn into glass capillary tubes. The protein in the glass capillary tubes was then subjected to a thermal ramp, and the absorbance at 350 nm and 330 nm was monitored. The first derivative of the ratio of 350 nm/330 nm was then plotted as a function of temperature by the onboard software, where the peak was defined as the melting temperature of the sample.

*AGR2 Protein Expression in PDAC and Normal Adjacent Tumor Tissue:* Human pancreatic tumor samples and their matched normal adjacent tissues from pancreatic resections were obtained from the Medical University of South Carolina Tissue Biorepository. These de-identified tissues were obtained from institutional surgical specimens under an IRB-approved protocol and stored at −80 °C prior to use. Samples were crushed in liquid nitrogen into a fine powder, and 30–100 mg were mixed in Radioimmunoprecipitation assay, or RIPA, buffer (R3792; TEKnova, Hollister, CA, USA) for protein lysis. Lysates were sonicated and centrifuged at 13,000× *g*, and supernatants were collected. Lysates were dialyzed, and then an additional RIPA buffer was added.

*AGR2 mRNA Analysis:* AGR2 mRNA expression levels were determined using The Cancer Genome Atlas (TCGA) database that was accessed on cBioPortal [26,27]. On cBioPortal, the AGR2 gene was queried for PDAC in the TCGA database, and descriptive statistics were performed. Gene Expression Profiling Interactive Analysis (GEPIA) [28] is an online tool that includes RNA sequencing data from several databases and can be used for interactive analysis of differential gene expression, profile plotting, survival analysis, and correlation analysis. GEPIA was used to analyze the differential RNA expression of AGR2 in pancreatic cancer in the TCGA compared to normal pancreatic tissue. Survival analysis with the Kaplan–Meier method was also conducted through GEPIA to determine the association of disease-free survival with high or low RNA expression of AGR2 in TCGA samples. We looked at publicly available datasets to explore percentiles at which there was a statistically significant decrement in survival. The cutoffs that we used were the 20th percentile or less for the low group and the 50th percentile or greater for the high group to allow for sufficient division between the groups to identify a statistical separation of survival.

## 3. Results

### 3.1. Characterization of AGR2 Expression in Normal Tissue and Primary Tumors

We first characterized AGR2 expression in primary PDAC biopsies using tissue microarray analysis (TMA). We found that AGR2 was undetectable in normal pancreas tissues but showed prominent staining in ~90% of PDAC samples (Figure 1A), which is consistent with previous reports [20,21,31]. Expression levels ranged from 5 to 70% of the tissue core (Figure 1B) and showed a significant increase with disease progression from stage I to stage II (Figure 1C). We further analyzed AGR2 expression in tumor and normal adjacent tissue from surgical samples acquired at our center. Similar to the TMA results, we observed a clear and statistically significant correlation between AGR2 expression in PDAC samples (Figure 1D, *p* = 0.0025, n = 24). With few exceptions, AGR2 expression was completely absent from normal adjacent pancreas tissue. Next, we compared *AGR2* mRNA expression in normal pancreas and pancreatic adenocarcinoma using the GEPIA online tool. PDAC mRNA expression data were obtained from the TCGA RNAseq database (n = 179), and normal pancreas mRNA expression data were obtained from the GTex RNAseq database (n = 177). *AGR2* mRNA levels in PDAC were significantly higher than in normal pancreas (*p* < 0.01; Figure 1E). The GEPIA online tool was then used to evaluate the association between *AGR2* mRNA expression and disease-free survival for pancreatic cancer patients. High AGR2 expression was defined as mRNA expression in the 50th percentile or higher, and low mRNA levels were defined as *AGR2* mRNA expression in the 20th percentile or lower. Disease-free survival was compared between high AGR2 (n = 107) and low AGR2 (n = 27) samples using the Kaplan–Meier method. High *AGR2* mRNA levels were associated with significantly lower disease-free survival (hazard ratio = 2.5; *p* = 0.016; Figure 1F). The findings indicate that AGR2 expression strongly correlates with PDAC disease progression and clinical outcomes and suggests that AGR2 is a candidate disease biomarker and potential therapeutic target.

### 3.2. Characterization of AGR2 Expression in PDAC Cell Models

We next evaluated and characterized the expression, subcellular localization, and molecular drivers of AGR2 expression in PDAC cell models. *AGR2* mRNA was detected in 7/8 PDAC cell lines (Figure 2A) and exhibited strong perinuclear staining (Figure 2B). Previous reports have shown AGR2 to be expressed on or associated with the external face of the cell membrane. We detected trace amounts of AGR2 in the cell surface fraction in cell surface biotinylation studies but were unable to detect it on the surface of cells by confocal imaging on unpermeabilized cells or by flow cytometry on live cells. While the results for cell surface localization were conflicting, we consistently observed AGR2 secreted into the cell culture media of cells expressing endogenous AGR2 or mouse PDAC cells from KPC (*Kras^G12D/+^*; *Trp53^R172H/+^*; *PDx-1-Cre*) mice that were engineered to overexpress wild-type AGR2 or a mutant form with a secretion signal that was highly secreted (Figure 2C). Loss of *SMAD4* is a hallmark of PDAC and other gastrointestinal tumor types, like colorectal cancer (CRC), and has been linked to AGR2 up-regulation. Indeed, we found that AGR2 expression was strongly and inversely correlated with SMAD4 expression in panels of PDAC and CRC cell lines (Figure 2D,E). By comparison, we observed no correlation with other PDI family members, such as PDIA1, suggesting that the linkage between SMAD4 and AGR2 was specific to AGR2 and not generalized to the entire enzyme family. We then proved that the association was beyond correlative and, in fact, causal, as SMAD4 knockdown in PANC-1 and HPAF-II cells led to a significant up-regulation of AGR2 (Figure 2F).

### 3.3. Characterization of AGR2 Expression in Normal Tissues

Previous studies have detected AGR2 mRNA in matched normal tissue from cancer patients in a select set of tissues [46,47], but a comprehensive examination of normal tissue expression has not been conducted to date. To address this, we performed an IHC analysis of AGR2 protein expression in TMAs from a comprehensive set of normal tissues acquired from multiple donors (N > 3). Tissue staining was evaluated qualitatively (Figure 3A) and scored quantitatively by a blinded individual using a 0–4 scoring system where 0 was no detectable staining, and 4 was >90% of the tissue staining positively for AGR2 (Figure 3B). We found the most intense staining in normal tissues of the GI tract from the stomach to the small intestine, colon, appendix, and rectum. Minimal but detectable levels of AGR2 were detected in several other organ systems, including the prostate, kidney, bladder, salivary gland, lung, and trachea. AGR2 was absent from the liver, ovary, brain, smooth and skeletal muscle tissues, as well as all hematological compartments (lymph nodes, bone marrow, thymus, and spleen).

### 3.4. AGR2 Enhances Adhesion and Migration of PDAC Cells

Previous studies have linked AGR2, specifically extracellular AGR2, to epithelial–mesenchymal transition (EMT) and increased migration and invasion of cancer cells [23,42,48]. To explore this hypothesis in our model systems, we used a traditional Boyden chamber assay to separate a lower cell culture chamber containing chemoattractant from an upper chamber containing cells. The two chambers were separated by a porous 8 μm membrane that migrating cells could pass through. Using this setup, we found that PANC-1 cells migrated toward a chemoattractant gradient composed of conditioned media (CM) collected from PANC-1 cells or fetal bovine serum. Most importantly, we found that the addition of recombinant AGR2 (rAGR2) to the cells in the top chamber significantly increased their ability to migrate (Figure 4A). Similar results were observed using invasion assays where PANC-1 cells were challenged to migrate through transwell membranes that were coated with extracellular matrix proteins (Figure 4B). We also found that coating with rAGR2 significantly increased the adhesion of PDAC cells to high-binding surface microplates. (Figure 4C). Next, we conducted PDAC spheroid formation studies to evaluate the impact of AGR2 expression on spheroid formation. We used cell lines derived from KPC mice, which spontaneously developed PDAC tumors [49]. Unlike human PDAC, KPC cells lack AGR2 expression, so we exogenously expressed human AGR2 to create an isogenic model of AGR2- and AGR2+ KPC cells (Figure 2C). Similar to what we observed for human PDAC cells, co-incubation of KPC cells with rAGR2 increased their baseline motility and migration toward a chemotactic gradient derived from KPC CM (Figure 4D). Genetic overexpression of AGR2 in KPC cells, like the addition of extracellular rAGR2 to KPC cultures, also enhanced baseline motility and migration toward CM (Figure 4E). We also evaluated the influence of AGR2 expression on the ability of KPC cells to form spheroids in culture. Interestingly, and somewhat unexpectedly, we found that KPC-AGR2 cells formed significantly smaller spheroids than KPC control cells with an average diameter of 40.3 ± 4 μm compared to 153 ± 11 μm for the AGR2-negative control KPC cells (Figure 4F). We noted that KPC-AGR2 cells behaved differently in these assays as they migrated from the center of the semi-solid Matrigel matrix plug to the periphery close to the interface with serum-rich media, where they formed many small spheroids. Altogether, these results demonstrate that AGR2, whether overexpressed or added to the extracellular environment, enhances the migratory phenotype of mouse and human PDAC cells.

### 3.5. Anti-AGR2 Antibody Discovery

Given the elevated expression of AGR2 in PDAC models and the apparent function of extracellular AGR2 in enhancing cell migration, adhesion, and invasion, we next set out to identify a high-affinity human anti-AGR2 antibody that could be used for the detection and neutralization of AGR2. To do this, we conducted phage display biopanning using a library of human single-chain variable fragments (scFv) arranged in VH-linker-VL format against the target AGR2 protein. After four rounds of phage selection, we confirmed the binding of four anti-AGR2 scFv clones with unique sequences by monoclonal phage ELISA and then by soluble scFv ELISA. We then cloned, expressed, and purified the four candidate scFvs in *E. coli* and confirmed specific binding to AGR2 by ELISA (Figure 5A). Of the top four clones, scFv-15 showed the strongest binding to AGR2 with an EC_50_ of 4.7 ± 0.52 μg/mL compared to scFv-1 (14.9 ± 0.88 μg/mL), scFv-19 (159 ± 20 μg/mL), and scFv-26 (204 ± 21 μg/mL). In addition to the highest binding affinity, scFv-15 exhibited the greatest thermodynamic stability with a melting temperature of 78.5 °C compared to 59.8 °C, 64.4 °C, and 70 °C for scFv-1, scFv-19, and scFv-26, respectively (Figure 5B). VH-VL sequences from scFv-15 were then cloned into a full human IgG1 backbone (MAb15), which showed similar binding affinity and specificity as scFv-15 (Figure 5C). MAb-15 showed similar binding to wild-type and mutant forms of AGR2 that were engineered to disrupt the CXXS catalytic redox core and the E60 residue within the homodimerization domain (Figure 5D) [50,51]. This demonstrates that MAb-15 binds to AGR2 independent of the redox or dimerization state of the protein. Therefore, these antibody discovery experiments successfully delivered a novel AGR2-specific monoclonal antibody with high thermodynamic stability and acceptable binding affinity as a lead candidate. Given these qualities, scFv-15/MAb-15 was carried forward into subsequent studies for further development.

### 3.6. Affinity Maturation of Anti-AGR2 MAb15

In order to improve the binding affinity of the anti-AGR2 MAb, we next conducted affinity maturation using the scFv-15 CDRs as a template. Mutant scFv-15 phage libraries of 10^9^–10^10^ complexity were constructed using error-prone PCR semi-rational approaches and then biopanned against biotinylated AGR2 protein [45]. Biopanning was performed in the presence of excess MAb-15 to select for clones with slower k*_off_* rates and, thus, higher binding affinity than the original. Multiple point mutations in the VH region were found to increase binding to AGR2 (Figure 6A). The S31R mutation improved AGR2 binding affinity by 13-fold (EC_50_ = 156 pM vs. 2030 pM for MAb-15), and the double mutation of S31R paired with A53F or A53Y showed the most significant improvement in affinity with 52.7-fold (EC_50_ = 38.5 pM) and 50.2-fold (EC_50_ = 40.4 pM) increases (Figure 6B,C). A T59P point mutation in the VH also improved affinity in combination with the A53F/Y mutations, although the three modifications together did not cooperate and actually appeared to antagonize the A53 variants. In addition, the T59P mutation imparted reduced thermodynamic stability with melting temperatures significantly lower than the S31R/A53F/Y clones (Figure 6D). Affinity maturation gave rise to antibody clones that were highly specific for AGR2. We tested binding to other PDI family members, such as PDIA1 and Thioredoxin (TXN), and used BSA as a control (Figure 6E,F). These studies found that S31R/A53F was 495-, 251-, and 975-fold more selective for AGR2, respectively, which was roughly 10 times more specific than the original MAb-15 clone. Similar results were observed using the S31R/A53F clone.

### 3.7. Anti-AGR2 MAbs Suppress AGR2 Activity

Anti-AGR2 antibodies were developed to generate a highly sensitive reagent for the detection of AGR2 as well as to create a therapeutic that neutralizes the pro-tumor function of extracellular AGR2. To determine the effects of novel AGR2 antibody clones, we tested their ability to block AGR2-mediated adhesion and migration in cell model systems. We found that the original clone, MAb-15, modestly inhibited AGR2-mediated adhesion, whereas the high-affinity clones (S31R/A53Y and S31R/A53F) more effectively and completely blocked adhesion at low nanomolar concentrations (Figure 7A). In cell migration experiments, we made similar observations. The high-affinity anti-AGR2 antibodies (S31R/A53Y) blocked the AGR2-enhanced migration of KPC cells toward a chemotactic gradient (Figure 7B). MAb-15 was less effective in these experiments, suggesting that its lower binding affinity limited its ability to block the phenotypic effects of AGR2. This was true in experiments where recombinant AGR2 was added to the extracellular environment of cells and when AGR2 was overexpressed in KPC cells. LY6/PLAUR Domain Containing 3 (LYPD3, a.k.a. C4.4A) was previously identified as a binding partner for extracellular AGR2 [52]. To confirm this interaction in our systems, we expressed recombinant versions of AGR2 and LYPD3 and evaluated in vitro binding by ELISA. We observed specific binding of LYPD3 to immobilized AGR2, thereby confirming a direct interaction between AGR2 and LYPD3 (Figure 7C). We further demonstrated that anti-AGR2 MAb clone S31R/A53Y blocked the interaction between AGR2 and LYPD3 in a dose-dependent manner (Figure 7D), further demonstrating the ability of this high-affinity antibody clone to disrupt the biological activity of AGR2. Taken together, these data show the ability of novel high-affinity anti-AGR2 antibodies developed in this study to bind and neutralize the pro-tumor activity of soluble AGR2.

## 4. Discussion

AGR2 is highly expressed in a variety of adenocarcinomas, including prostate [53], breast [54], and lung cancers [55] and others. While its expression is linked to oncogenesis, its precise role in promoting and/or maintaining the malignant phenotype is not fully understood. In this study, we used a variety of methodologies that demonstrated high levels of AGR2 expression in PDAC patient biopsies and cell models. The normal pancreas was completely devoid of AGR2 expression, as were most other normal tissues, with the exception being tissues of the stomach and proximal and distal GI tracts. With regard to subcellular localization, we found that AGR2 was highly expressed in the intracellular compartment of the ER and secreted into the extracellular space. The ER localization is expected as AGR2 expresses a KTEL motif, which is an ER retrieval signal that binds to the KDEL receptor (KDELR). Despite this, previous reports have shown AGR2’s association with the external face of the plasma membrane [31,37]. In our study, however, we were unable to detect appreciable levels of AGR2 associated with the cell membrane by flow cytometry or by fluorescence microscopy in unpermeabilized AGR2-positive PDAC cells. Our experiments incorporated multiple human and mouse PDAC cell lines with exogenous AGR2 overexpression, none of which presented AGR2 on the cell surface. AGR2 would not be the first ER-resident protein lacking a transmembrane domain to be found on the cell surface of cancer cells. For example, glucose-regulated protein 78 (GRP78/BiP), which is an ER resident chaperone, translocates from the ER to the cell surface following certain stimuli [56,57]. GRP78 has been targeted for immunotherapy using bispecific T-cell redirecting antibodies and chimeric antigen receptor T-cell therapies (CAR-T) that require the formation of an immune synapse and necessitate a cell surface-expressed ligand [58,59,60]. Additional studies may be warranted to explore the targetability of cell surface AGR2, although our data cast doubt on its utility as a target for anti-cancer T-cell therapy.

Our study highlights the pro-tumor function of extracellular AGR2 (not cell surface-localized) and supports the use of agents that bind and neutralize AGR2 function. Our results show that AGR2 modulates the mobility of PDAC cells. We demonstrated this in the experiments, where recombinant AGR2 was added directly to cell culture media and other studies with novel high-affinity MAbs that neutralized the extracellular function of AGR2. One of the unanswered questions from this work is why AGR2, an ER resident protein, is secreted from cells. It has been suggested that the atypical KTEL motif of AGR2 has a low binding affinity for the KDELR, leading to loss of ER localization and secretion of AGR2 [61], although contradicting studies have challenged this hypothesis [62]. Questions remain about this part of the mechanism, but what is clear from our data is that AGR2 regulates cell adhesion, migration, and invasion. AGR2 also has a number of characteristics that overlap with classical members of the EMT pathway. For example, previous studies have shown that AGR2 expression was regulated by TGF-beta, a strong inducer of EMT [42,63], and SMAD4, an important downstream effector in the TGF-beta signaling pathway [26]. Consistent with this, our data show causality between AGR2 expression and loss of SMAD4 in PDAC cells. SMAD4 deficiency is coincidentally one of the four major mutations in PDAC, suggesting that AGR2 up-regulation may play a role in PDAC initiation and progression downstream of SMAD4 loss. Altogether, these observations suggest that AGR2 plays a role in the progression of PDAC toward the metastatic phenotype, although, to our knowledge, a causal relationship between AGR2 and the formation of metastatic tumors has not been reported in the literature in PDAC or any other solid tumor model.

The discovery of a series of high-affinity anti-AGR2 MAbs (<50 pM EC_50_) is a key innovation in our study. These molecules have therapeutic potential in PDAC as well as other AGR2-expressing solid tumors. An important question around the therapeutic utility is whether AGR2 has the same function in normal tissues as in tumors. It is also not clear whether AGR2 secretion is a cancer cell-specific phenomenon or if it is also secreted from the limited number of normal tissues that express it under physiological conditions (e.g., colon). In normal intestinal epithelial cells of the mouse, AGR2 was essential for the production and secretion of proteins that form the intestinal mucus barrier [64]. This role was found to protect mice from colitis but was solely an intracellular function of AGR2. Our study and those of others show a prominent role of AGR2, and extracellular AGR2 in particular, in cancer cell migration, although if and how this motility function is important for normal colon cells is unknown. This question has significant therapeutic implications for our study. If AGR2 secretion and impact on cell migration is limited to cancer cells, then the therapeutic MAbs we present herein potentially target a cancer-specific mechanism that should be given high priority in PDAC.

## Figures and Tables

**Figure 1 antibodies-13-00101-f001:**
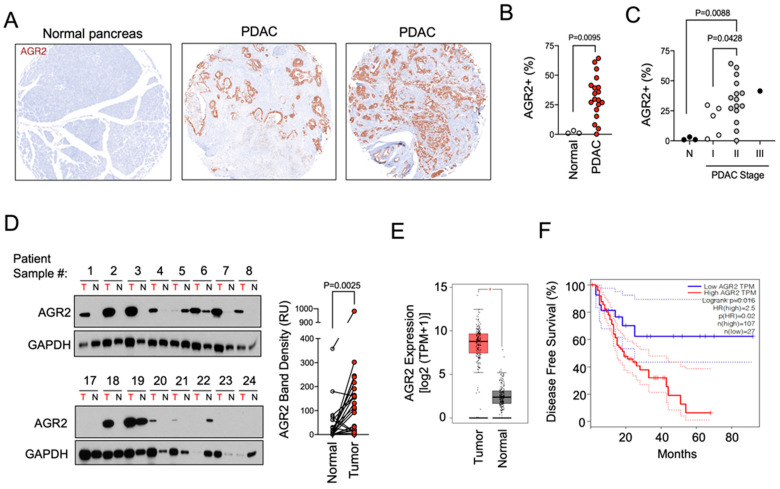
AGR2 is overexpressed in PDAC but not normal pancreas tissue. (**A**) Immunohistochemical staining for AGR2 was conducted on tissue microarray (TMA) slides constructed of tissue cores from normal PDAC tissues. Representative images of tissue cores from normal and two PDAC tissue samples are shown. (**B**) AGR2 staining was quantified in the TMAs analyzed in (**A**). Data are expressed as percent of nucleated cells staining positive for AGR2. (**C**) AGR2 staining analysis from (**B**) was segmented by PDAC stage. Statistical significance between groups was determined using a Student’s *t*-test. (**D**) Tumor and normal adjacent tissue from PDAC patients were harvested and analyzed by Western blotting for AGR2 expression. AGR2 blots are shown along with quantification of band densitometry between matched tumor (T) and normal adjacent (N) tissue samples. Statistical significance between the two groups was determined using a Student’s *t*-test (n = 24). (**E**) The GEPIA online tool was used to compare the mRNA gene expression of *AGR2* in the TCGA database in pancreatic cancer (n = 179) with the mRNA expression of AGR2 in normal pancreas samples from the GTex database (n = 177). *AGR2* mRNA levels were significantly higher in pancreatic adenocarcinoma samples compared to normal pancreas samples (*p* < 0.01). (**F**) GEPIA was used to compare disease-free survival between patients with high *AGR2* mRNA levels (n = 107) and low *AGR2* mRNA levels (n = 27) using the Kaplan–Meier method. High *AGR2* expression was associated with significantly worse disease-free survival (hazard ratio = 2.5; *p* = 0.016).

**Figure 2 antibodies-13-00101-f002:**
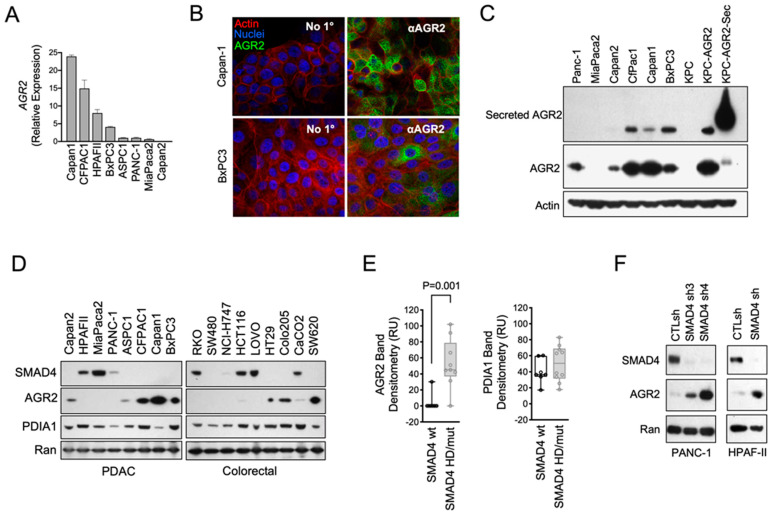
Intracellular and extracellular AGR2 levels are up-regulated in PDAC and colorectal cancer cell models in an SMAD4-dependent manner. (**A**) AGR2 gene expression was measured by RT-qPCR in the indicated PDAC cell lines. Expression data were normalized to a GAPDH internal control and expressed relative to PANC-1 levels. (**B**) Capan1 and BxPC3 cells were fixed and permeabilized, stained for AGR2 expression, and imaged by confocal microscopy. Strong perinuclear staining of the endoplasmic reticulum can be observed. (**C**) Western blots for secreted AGR2 (from cell culture supernatant) and whole cell AGR2 are shown by Western blotting for the indicated cell lines. KPC mouse PDAC cells are included as a no-expression control or overexpressed with wild-type AGR2 (KPC-AGR2) or with a mutated AGR2, expressing a secretion signal (KPC-AGR2-Sec) as secreted AGR2 positive control. (**D**) Western blots are shown for the indicated PDAC and colorectal cancer cell lines. (**E**) AGR2 and PDIA1 band densitometry from the cell lines in (**D**) were quantified. A statistically significant difference in AGR2 expression between SMAD4 wild-type and SMAD4 deficient cells was determined using a Student’s *t*-test (n = 7 SMAD4 wild-type, n = 9 SMAD4 deficient). (**F**) SMAD4 was silenced using multiple targeted shRNA sequences in the indicated PDAC cell lines. Western blots are shown.

**Figure 3 antibodies-13-00101-f003:**
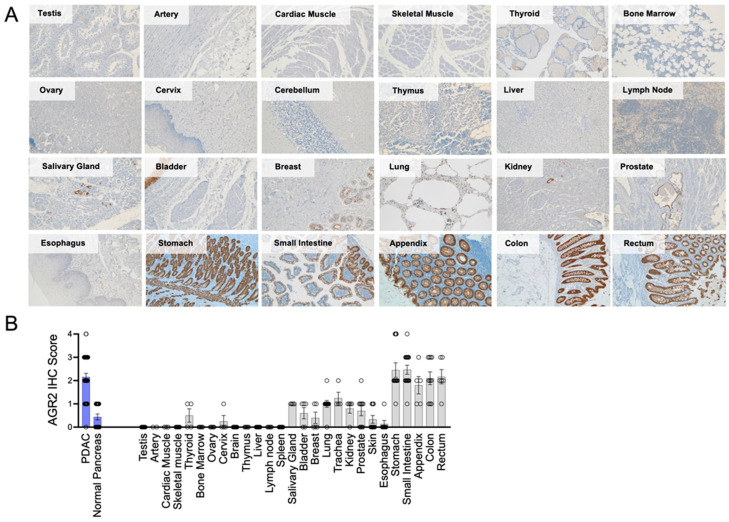
Characterization of AGR2 expression in normal human tissues. (**A**) Immunohistochemical analysis of AGR2 expression was conducted on TMAs constructed of cores from the indicated normal human tissues. Representative images from AGR2 stained cores are shown. (**B**) An AGR2 staining score (0–4) was assigned by a blinded investigator to each tissue core analyzed in (**A**).

**Figure 4 antibodies-13-00101-f004:**
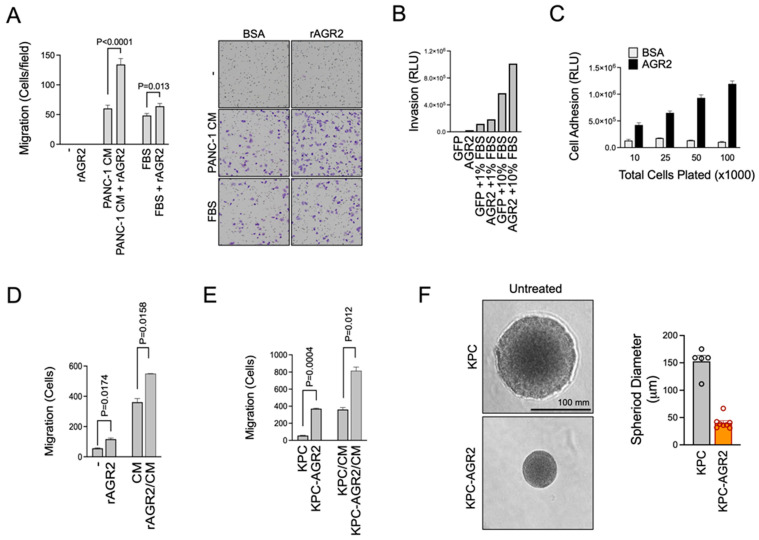
AGR2 enhances migration and invasion of PDAC cells. (**A**) PANC-1 cells were used in migration assays using a Boyden chamber design with 8 μm porous membrane. PANC-1 cells were added to the top chamber in the presence or absence of rAGR2 (500 nM) and allowed to migrate toward a chemoattractant gradient composed of PANC-1 conditioned media (CM) or 10% FBS for 16 h. Quantification of migrated cells is shown on the left, with representative images of Giemsa-stained membranes on the right. (**B**) Experiments analogous to those described in (**A**) were conducted with the porous membrane coated with extracellular matrix proteins to simulate invasion and migration. Cells that invaded and migrated through the porous membrane toward a chemoattractant gradient of 10% FBS were measured by detaching cells from the underside of the invasion chamber and measuring viable cells with the CellTiter-Glo assay system coupled with a luminescence imager. (**C**) High protein binding ELISA plates were coated with BSA (control) or rAGR2. PANC-1 cells were plated at the indicated cell densities and allowed to adhere to plates for 1 h. Non-adhered cells were washed away, and adherent cells were subsequently quantified using the CellTiter-Glo assay. (**D**) KPC mouse PDAC cells were used in migration experiments in a Boyden chamber-like design similar to those described in (**A**). Cells were plated in the top chamber with or without rAGR2 (500 nM) and allowed to migrate toward a chemoattractant gradient composed of KPC CM for 6 h. Cell migration data are shown. Statistical significance was determined by Student’s *t*-test. (**E**) KPC or AGR2-expressing KPC cells (KPC-AGR2) were allowed to migrate toward a KPC CM gradient for 6 h. Cell migration data are shown. Statistical significance was determined by Student’s *t*-test. (**F**) PDAC spheroids were grown in 3D cultures. Representative images of spheroids from KPC or KPC-AGR2 cells are shown on the left. Spheroid diameter was quantified and shown on the right.

**Figure 5 antibodies-13-00101-f005:**
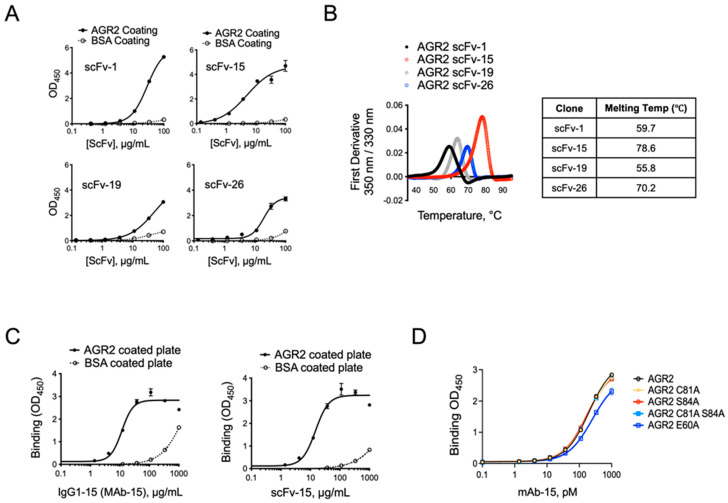
Discovery and validation of anti-AGR2 scFvs. (**A**) Anti-AGR2 scFv clones 1, 15, 19, and 26 were produced, purified, and evaluated for binding to AGR2-coated or BSA-coated (control) plates. ELISA data are shown. (**B**) Thermodynamic stability studies were conducted using the indicated anti-AGR2 scFv clones. Raw data (top) and extrapolated melting temperature data (bottom table) are shown. (**C**) The VH-VL sequence of anti-AGR2 clone scFv-15 was cloned into a fully human IgG1 backbone. Binding to AGR2-coated or BSA-coated (control) plates is shown for the full IgG1 (MAb-15, left) and the original scFv format (right). (**D**) MAb-15 binding to plates coated with the indicated AGR2 mutations is shown by ELISA.

**Figure 6 antibodies-13-00101-f006:**
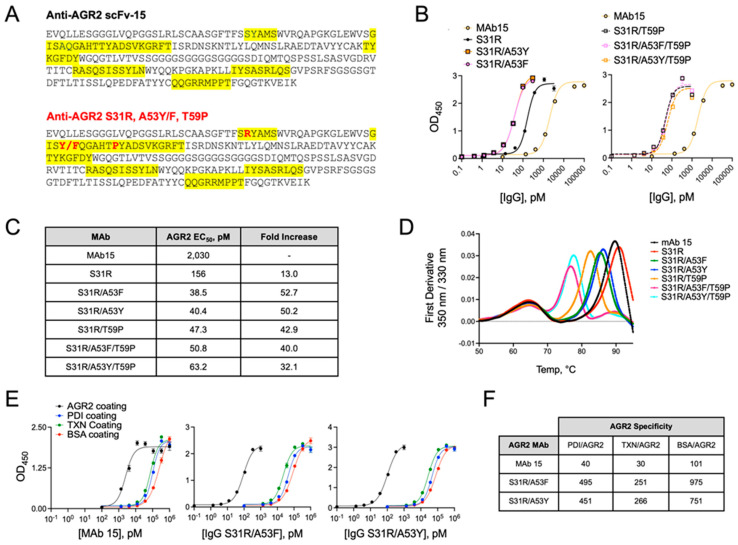
Affinity maturation of MAb-15 produces high-affinity AGR2 antibodies. (**A**) VH-VL amino acid sequence is shown for the original anti-AGR2 clone scFv-15 (top) along with the sequence, containing mutation sites within complementarity determining regions (CDRs, bottom) that enhance binding to AGR2. CDRs are highlighted in yellow, and mutated residues are shown in bold red font. (**B**) The relative binding to AGR2-coated plates is shown for the original anti-AGR2 antibody, MAb-15, along with the indicated affinity-enhanced clones. ELISA data are shown. (**C**) EC_50_ values were extrapolated from the ELISA binding curves in (**B**). Those values are shown along with the fold increase in binding over the original MAb-15 clone. (**D**) Thermodynamic stability studies were conducted using the indicated anti-AGR2 IgGs. Raw data are shown. (**E**) Binding experiments were conducted using plates coated with AGR2 or the PDI family member, PDI (PDIA1), thioredoxin (TXN), or BSA (control). ELISA data for MAb-15 and the two indicated affinity-matured clones are shown. (**F**) EC_50_ values were extrapolated from the binding curves in (**E**) and expressed relative to AGR2 binding as an expression of the specificity of each clone for AGR2 compared to the other proteins. Relative values are shown, with higher values indicating the greatest level of specificity.

**Figure 7 antibodies-13-00101-f007:**
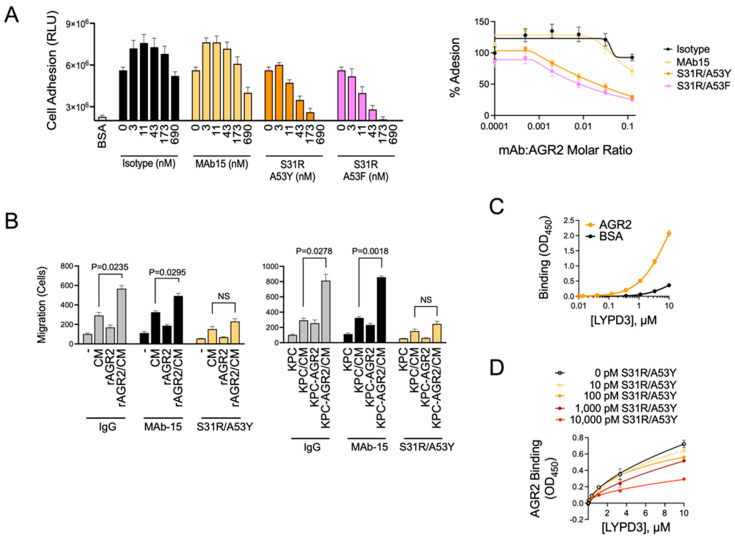
High-affinity anti-AGR2 MAbs inhibit AGR2-mediated cell adhesion, migration, and binding to LYPD3. (**A**) ELISA plates were coated with BSA or AGR2, and PANC-1 cells were allowed to adhere for 1 h in the absence (“0”) or presence of the indicated anti-AGR2 MAbs. Non-adherent cells were washed away, and adherent cells were quantified using the Cell Titer-Glo assay system. Cell adhesion data are shown. (**B**) KPC cells were used in migration assays using a Boyden chamber design with 8 μm porous membrane. (Left) KPC cells were added to the top chamber in the presence or absence of rAGR2 (500 nM) and allowed to migrate toward a chemoattractant gradient composed of KPC-conditioned media (CM) for 6 h. Quantification of migrated cells is shown. (Right) KPC or KPC cells with AGR2 overexpression were added to the top chamber and allowed to migrate toward a chemoattractant gradient composed of KPC-conditioned media (CM) for 6 h. Quantification of migrated cells is shown. Statistical significance was determined using a Student’s *t*-test (n = 3). (**C**) LYPD3 binding to AGR2-coated or BSA-coated (control) plates was analyzed. ELISA data are shown. (**D**) LYPD3 binding to AGR2-coated plates was measured in the presence of the indicated concentrations of the affinity-mature anti-AGR2 MAb S31R/A53Y. ELISA data are shown.

## Data Availability

The original contributions presented in this study are included in the article. Further inquiries can be directed to the corresponding author(s).

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
