# Peer review of "A High-Affinity Monoclonal Antibody Against the Pancreatic Ductal Adenocarcinoma Target, Anterior Gradient-2 (AGR2/PDIA17)"

_2073-4468, 2024, doi:10.3390/antib13040101_

Round 1

Reviewer 1 Report

Comments and Suggestions for Authors

In this paper the authors show a lot of nice data that validate AGR2 as a potential therapeutic target in pancreatic ductal 20 adenocarcinoma (PDAC), and then describe some novel AGR2 scFv antibodies and reformatted full-size antibodies that show stong binding affinity to this target.  These antibodies show the ability to reduce cellular motility in cell models compared to controls which suggests potential as a therapeutic against PDAC. 

Minor typos:

Line 173  change “to enriched phage clones”  to  “ to enrich phage clones”

Line 177 “The candidate scFv mutants were formatted into human IgG1 and further re-combinantly expressed and purified by 293 cells and measured for affinity kinetics by Octet BLI.”  I can’t see any BLI data for the reformatted antibodies.  Add this data or omit this line.

Line 402 “(Ref)” missing annotation

Line 540 “Koff” to “koff” (lower case k)

Line 546 remove semicolon at the end of the line

Line 563 Change “Thos values” to “Those values”

Figures:

Figures are finalized?  They are very low resolution and need to be improved for legibility.

Fig 3 – order the histology photos in A and bar graph data in B the same

Fig 5A – put the data in the same panel for comparison, same with Fig 5D

My main concern for the paper is that the poor BLI experiment with scFv-15 in Figure 5C is used as a benchmark to pM binding. It is an outlier with respect to the data quality in the paper and detracts from the other data presented.  No repeats were done and the 1:1 fit isn’t shown which will be poor.  There is a large biphasic change in the dissociation phase which could be do to aggregate or multimers due to the E. coli preparation not being further purified.  UPLC-SEC should be performed to verify the homogeneity of scFv-15 (and the other variants generated).  The authors may have more success with capturing the scFv and flowing the AGR2, although often cleaved preparations are not 100% devoid of His and will still show some anti-His capture.  AGR2 may still require SEC purification to remove multimers or aggregate for quality label-free kinetics. I think the authors will find it easier to capture the reformatted mAb and flow AGR2 to generate some nice kinetics for the the mAb-15 and the affinity matured variants. Generally the mAbs will behave much better than scFvs.  It would be desirable to understand the binding kinetics of the mAbs against AGR2 as a soluble target when as potential therapeutic candidates they will be acting as a ligand trap.  At minimum remove the BLI data entirely and just refer to the EC50 values from the ELISA assays or redo the BLI as described (remove the reference to pM binding affinity in line 32, for instance, and change to EC50s).

Author Response

Comments and Suggestions for Authors

In this paper the authors show a lot of nice data that validate AGR2 as a potential therapeutic target in pancreatic ductal 20 adenocarcinoma (PDAC), and then describe some novel AGR2 scFv antibodies and reformatted full-size antibodies that show stong binding affinity to this target.  These antibodies show the ability to reduce cellular motility in cell models compared to controls which suggests potential as a therapeutic against PDAC. 

Thank you for your time and thoughtful review. We have addressed your critiques and recommended changes as stated below and in the revised manuscript.

Minor typos:

Line 173  change “to enriched phage clones”  to  “ to enrich phage clones” Change made

Line 177 “The candidate scFv mutants were formatted into human IgG1 and further re-combinantly expressed and purified by 293 cells and measured for affinity kinetics by Octet BLI.”  I can’t see any BLI data for the reformatted antibodies.  Add this data or omit this line. This sentence was corrected to read: The candidate scFv mutants were formatted into human IgG1 and further recombinantly expressed and purified by 293 cells and measured for affinity kinetics by ELISA.”

Line 402 “(Ref)” missing annotation This was removed

Line 540 “Koff” to “koff” (lower case k) Change made

Line 546 remove semicolon at the end of the line Change made

Line 563 Change “Thos values” to “Those values” Change made

Figures:

Figures are finalized?  They are very low resolution and need to be improved for legibility. Higher resolution images will be submitted.

Fig 3 – order the histology photos in A and bar graph data in B the same Change made

Fig 5A – put the data in the same panel for comparison, same with Fig 5D We kept these data separated because the overlaying of all the on-target (AGR2) and non-specific (BSA) binding curves when combined appear cluttered. We made sure to keep the axis scales the same so that accurate comparisons could be made.

My main concern for the paper is that the poor BLI experiment with scFv-15 in Figure 5C is used as a benchmark to pM binding. It is an outlier with respect to the data quality in the paper and detracts from the other data presented.  No repeats were done and the 1:1 fit isn’t shown which will be poor.  There is a large biphasic change in the dissociation phase which could be do to aggregate or multimers due to the E. coli preparation not being further purified.  UPLC-SEC should be performed to verify the homogeneity of scFv-15 (and the other variants generated).  The authors may have more success with capturing the scFv and flowing the AGR2, although often cleaved preparations are not 100% devoid of His and will still show some anti-His capture.  AGR2 may still require SEC purification to remove multimers or aggregate for quality label-free kinetics. I think the authors will find it easier to capture the reformatted mAb and flow AGR2 to generate some nice kinetics for the the mAb-15 and the affinity matured variants. Generally the mAbs will behave much better than scFvs.  It would be desirable to understand the binding kinetics of the mAbs against AGR2 as a soluble target when as potential therapeutic candidates they will be acting as a ligand trap.  At minimum remove the BLI data entirely and just refer to the EC50 values from the ELISA assays or redo the BLI as described (remove the reference to pM binding affinity in line 32, for instance, and change to EC50s). We have removed the Octet-BLI per your suggestion and adjusted the text in the Results, Materials and Methods, and Figure Legends to reflect the change.

Reviewer 2 Report

Comments and Suggestions for Authors

In this manuscript, the authors studied the function and expression of AGR2 in PDAC and found that AGR2 is highly expressed in PDAC, and AGR2 level is correlated with the disease stage, which makes AGR2 a promising signaling marker for PDAC. Furthermore, the authors over-expressed the recombinant version of AGR2 and discovered that AGR2 is responsible for the adhesion, motility, and invasion in both human and mouse PDAC cell models. On top of the above findings, an iterated phage display screening was conducted to select the AGR2-specific scFv variants with the highest binding affinity to produce the monoclonal antibody (MAb). The newly discovered MAbs of AGR2 were then tested with the existing AGR2 binding partner LYPD3 on the inhibition effects of AGR2-mediated cell adhesion, migration, and LYPD3-binding. The experimental results showed outstanding performance and therefore suggests novel, high affinity, fully human anti-AGR2 MAbs with a potential to neutralize the pro-tumor effects of AGR2 in PDAC. The manuscript demonstrates a logical experiment design and the interpretation of results is reasonable. The language is also of good quality to elucidate the authors’ ideas clearly without ambiguity. I only have a few points of view that I would like the authors to respond to:

1.      In the lane 135, “expi2393F cells” should be “expi293F cells”.

2.      In the phage biopanning session, I am a little concerned that since the phage display screening was against a recombinant human 6X-His tagged AGR2 protein, will the His tag interfere with the screening? Is the His-tag on the N- or C-terminus to the AGR2 gene?

3.      In the affinity maturation of anti-AGR2 MAb15, I noticed that the important mutations are mostly on the S31, A53, and T59 location. The phage library is of 10^9-10^10 complexity so if using NNK codon to generate all 20 amino acids, the number of randomized amino acids should be 7 or 8. I am wondering what the other randomized amino acid residues are and what is the rationale for choosing all of these amino acid residues to randomize (including S31, A53, and T59). And in the four rounds of screening, does the screening condition get more and more strict (e.g. slower and slower Koff)?

4.      For the selected S31R, A53Y/F, and T59P mutations that drastically increased the binding affinity of MAb-15, do the authors care to venture a plausible mechanism on the amino acids level of how those mutations enhance the binding and where those mutations bind to on the AGR2?

5.      On the lane 578, “S31RA53F” should be “S31R/A53F”.

Author Response

In this manuscript, the authors studied the function and expression of AGR2 in PDAC and found that AGR2 is highly expressed in PDAC, and AGR2 level is correlated with the disease stage, which makes AGR2 a promising signaling marker for PDAC. Furthermore, the authors over-expressed the recombinant version of AGR2 and discovered that AGR2 is responsible for the adhesion, motility, and invasion in both human and mouse PDAC cell models. On top of the above findings, an iterated phage display screening was conducted to select the AGR2-specific scFv variants with the highest binding affinity to produce the monoclonal antibody (MAb). The newly discovered MAbs of AGR2 were then tested with the existing AGR2 binding partner LYPD3 on the inhibition effects of AGR2-mediated cell adhesion, migration, and LYPD3-binding. The experimental results showed outstanding performance and therefore suggests novel, high affinity, fully human anti-AGR2 MAbs with a potential to neutralize the pro-tumor effects of AGR2 in PDAC. The manuscript demonstrates a logical experiment design and the interpretation of results is reasonable. The language is also of good quality to elucidate the authors’ ideas clearly without ambiguity. I only have a few points of view that I would like the authors to respond to:

Thank you for your time and thoughtful review. We have addressed your critiques and recommended changes as stated below and in the revised manuscript.

  1. In the lane 135, “expi2393F cells” should be “expi293F cells”. Change made
  2. In the phage biopanning session, I am a little concerned that since the phage display screening was against a recombinant human 6X-His tagged AGR2 protein, will the His tag interfere with the screening? Is the His-tag on the N- or C-terminus to the AGR2 gene? We were confident in the specificity of binding to AGR2 because of data in Figure 6D that shows >100-fold better binding of MAb15 to AGR2 over other PDIs, which also express 6X His tags. The affinity mature clones showed greater than 1000-fold better binding as well. In other studies (not yet published) we also created a cell line expressing a transmembrane anchored form of AGR2 (no His tag) and showed binding of our antibody candidates further giving confidence of binding to AGR2 rather than the affinity tag. We used a C-terminal His tag for the recombinant AGR2 protein.
  3. In the affinity maturation of anti-AGR2 MAb15, I noticed that the important mutations are mostly on the S31, A53, and T59 location. The phage library is of 10^9-10^10 complexity so if using NNK codon to generate all 20 amino acids, the number of randomized amino acids should be 7 or 8. I am wondering what the other randomized amino acid residues are and what is the rationale for choosing all of these amino acid residues to randomize (including S31, A53, and T59). And in the four rounds of screening, does the screening condition get more and more strict (e.g. slower and slower Koff)? For the affinity maturation, we only focused on the most relevant amino acid changes that were relevant to improving binding to AGR2. There was not any rationale for selecting those 3 amino acids to mutate, as they were found through random screening. With regards to the screening conditions getting more strict, you are correct, the four rounds of screening were conducted in a way to select for clones with a slower koff. This was accomplished by performing the screening in the presence of excess amounts of the original antibody clone MAb-15. This is discussed in the Materials and Methods section in the “Antibody Affinity Maturation” section (lines 166-179).
  4. For the selected S31R, A53Y/F, and T59P mutations that drastically increased the binding affinity of MAb-15, do the authors care to venture a plausible mechanism on the amino acids level of how those mutations enhance the binding and where those mutations bind to on the AGR2? Thank you for the question. It would be interesting to know but difficult and purely speculative to venture a guess without structural data in-hand. We would need crystallography studies to show how the mutations affect binding.
  5. On the lane 578, “S31RA53F” should be “S31R/A53F”. Change made